**communications** engineering

# A framework for small satellite deployable structures and how to deploy them reliably
Jonathan Sauder [1] ✉, Christine Gebara[1], Narravula Harshavardhan Reddy [1] & Carlos J. García-Mora[1,2]

Because of the miniaturization of small satellites, most of them have deployables to expand effective areas. However, Small Satellites are not only required to miniaturize systems, but often have a reduced budget, timeline, and employ teams with less experience. The goal of this paper is to provide a starting point for those new to deloyables, and working on small satellites, to understand the approaches available for deployable mechanisms and provide design practices which can improve success rates. To do so, this paper develops a framework for small satellite deployable structures, categorizing them into distinct deployment stages. It investigates the approaches that can be utilized for each stage, focusing on the stow, restrain, actuate, and locate stages. This review paper discusses the advantages and disadvantages of each approach, supported by examples provided in the references. It then highlights best practices for deployable mechanisms, and describes key challenges and future directions. By offering a comprehensive analysis of small deployable systems, this paper aims to guide engineers and researchers in implementing successful design practices for small satellite deployable structures.

Initially limited to Low Earth Orbit (LEO)[1], Small Satellites have expanded their reach throughout the solar system with advancements like Mars Cube One (MarCO) in 2018[2] and are performing real science[3]. However, due to the compact size of Small Satellites, there is often a need to incorporate deployable mechanical systems to expand their functionality once in space. Research on numerous CubeSat missions reveals that over 90% of them employ deployable structures actuated by mechanical systems[4], such as antennas, solar panels, and instrument booms.

Designing mechanisms for Small Satellites presents a serious challenge, as simply scaling down mechanisms designed for larger spacecraft is not feasible. Instead, these mechanisms must be specifically tailored to reduce complexity, making good mechanical design practices essential. Complexity must not only be reduced to miniaturize components, but also because small satellites teams are often less experienced, have access to limited facilities, are provided limited budgets and experience accelerated schedules to make a shared launch opportunity. Therefore, being able to select a low complexity, robust architecture quickly, by following robust design practices will help with delivering deployables that are short on time and budget. Further, the recent professionalization of small satellites has reduced the acceptance of failure, further increasing the importance of selecting robust design through good design practices.

The goal of this paper is to provide a starting point for those new to small satellite deployables to understand the approaches available for deployable mechanisms and design practices they should follow. This paper elucidates design practices by first developing a framework of deployable structures, which starts with the stages of deployment and then discusses approaches for each stage. Then, design practices to ensure successful implementation are discussed, starting with general practices which are applicable to all deployables, and then moving into practices which are specific to certain approaches. Finally, we explore current challenges facing small satellite deployables, and future directions through new opportunities. This paper differentiates itself from prior review papers[4–8] through the creation of a framework which aids in understanding key approaches to deployables instead of dividing deployables into their functional applications (e.g., antennas, booms, etc.). It also specifically focuses on recent developments related to small satellites, versus many prior works that have focused on larger satellites[7,8].

## Defining a framework for small satellite deployables

To create a framework, deployment stages must first be defined. The deployment process starts with stowing the system. Stowing not only finds a way to shrink the system into a compact volume but also plays a key role in how the deployment sequence occurs. After the system has been stowed, it must be restrained to prevent accidental deployment prior to the intended time and to prevent it from moving due to launch loads.

[1]Jet Propulsion Laboratory, California Institute of Technology, Pasadena, CA, USA. [2]Department of Mechanical Engineering and Manufacturing, Universidad de Sevilla, Seville, Spain. ✉e-mail: jsauder@jpl.nasa.gov

After launch and when deployment is desired, the next step is to deploy. The deployment itself is divided into three stages, initial unstow, guide, and actuate. Some deployment systems have a mechanism or device which causes initial unstow or first motion, like kickoff springs. The next step, deployment guidance, has two substages, the first controlling how the deployable moves geometrically and the second how the speed of deployment is controlled. To drive the deployment motion, an actuator is required to apply the force to result in the deployment.

At the end of deployment, a locating feature(s) may be required to stop the deployment and determine its end state. Sometimes a latch is required to secure the deployable.

In summary, the stages of deployment are identified as stow, restrain, initial unstow, guide geometry, guide speed, actuate, locate, and latch. To keep the paper concise, it was decided to focus on four of the eight stages; Stow, Restrain, Actuate and Locate. While all the stages are important, these four stages especially drive deployable design and function.

## Approaches for deployment stages

The framework is expanded from stages to approaches for each of the four key stages. Many of these approaches were preliminarily identified by exploring a history of CubeSat Deployables[4] and by reviewing deployment mechanisms for small satellites. Beyond the text description below, a visual representation of each approach is provided in Fig. 1, organized in the same order as the section headings below.

## Stow approaches

The approach used to stow a deployable before launch often impacts approaches for later stages, as certain stow approaches are only compatible with some restraint or actuation approaches.

### Parallel fold(s)

Parallel folds are the most common approach, where the deployable is folded on a single axis, or set of parallel axes. While this approach is most used to fold flat panels, it is also used in booms and dipole antennas. Examples include solar panels[9–11]. Historically, many deployable structures forming large planar surfaces consisted of rigid panels connected by hinges. Folding the panels can be done through Z-folding or tri-folding[12]. Z-folding involves alternating mountain and valley folds. In tri-folding, two panels are folded in the same direction onto a central panel. This approach was used in several CubeSats including NASA Starling[13]. Both these folding approaches were used together in the TacSat thin film solar arrays[14].

Advantages: Z-folding can't jam as the panels do not intersect. It can be made into a radial fold like a paper fan or the Ultraflex solar arrays[15]. Tri-folding enables a symmetric deployed state. It can also be easily combined with Z-folding to increase the surface area in deployed state (while not increasing the cantilever length).

Disadvantages: Parallel fold panels consume larger surface areas on a satellite when compared to other stowing methods, as the panels have fewer fold lines. This means that they can block instruments, telemetry, or body mounted solar panels from operating on the surfaces they cover prior to deployment. Also, thickness of the stack increases with increase in desired surface area. For Z-folding, the length of cantilever increases with increasing number of folds. To utilize the 1 degree-of-freedom (dof) deployment, a deployable boom may be needed which introduces additional complexity. Tri-folding has multiple dofs which means jamming can occur. Since two panels are folded onto the same central panel, the two folds cannot be identical. One fold needs a thicker crease or more offset at the hinge.

Failure Mechanisms: Because of the simplicity of parallel folds, failure mechanisms are often a part of the other deployment stages, such as actuating hinges. For example, the parallel fold solar panels on the Momentus Vigoride 3 failed to deploy due to a pin in a hold down bracket failing to release[16]. Tri-folds can fail due to jamming.

### Orthogonal and parallel folds

Orthogonal/parallel folds occur when the deployable is folded multiple times, such that there are several deployment axes between the deployable and the satellite. Fold lines are not only in parallel axes but may occur at 90-degree angles to each other. Examples include the MarCO high gain antenna and solar panels[2], and the Prometheus Block 2 solar panels[10].

Advantages: Orthogonal fold panels are also simple and tend to stow more compactly than parallel folds. Further, deployment geometries can be more complicated, when they deploy across multiple axes, meaning unique starting and ending positions are possible.

Disadvantages: Sequencing of the deployment of each axis must be phased to stop self-intersection and jamming of various deployable panels. In an early version of MarCO, the panels deployed towards the spacecraft, creating a jamming risk[17]. With increase in number of folds, width of creases increases resulting in bigger gaps or loss of usable surface area.

Failure Mechanisms: One key risk of these panels is jamming due to sequencing, when a panel may deploy in a way that prevents other panels from moving.

### Origami/Kirigami Folds

Origami folds occur when the system is not folded in a set of simple folds, and the fold lines do not necessarily occur at 90-degree angles to each other. Folds may occur at all angles. Origami folding is where the sheet is continuous when unfolded, whereas kirigami folding has cuts in the sheets. Examples include the origami flasher pattern[18], Miura-Ori folded solar sails[19,20], antennas[21]; kirigami Z-folded solar array by Deployables Cubed[22], and the Caltech Space Solar Power Project[23].

Advantages: Origami/Kirigami folds can offer better packaging efficiency compared to Parallel or Parallel/Orthogonal folds, resulting in a more compact volume. Furthermore, the deployed system can be designed to have a single degree of freedom, requiring only one actuator.

Disadvantages: Origami requires a high level of expertise and analysis. It is not straightforward to develop complex folding patterns. Further, unique issues arise like rigid foldability[24] and self-intersection[25]. Due to the cuts or gaps in Kirigami-folded structures, snapping between adjacent panels or segments could occur leading to material failure and/or unsuccessful deployment. The panels could also be caught in the deployment mechanism.

Failure Mechanisms: Multiple degrees of freedom in an origami deployment can result in the system jamming during deployment due to intersection[26], especially if panels deploy in the wrong sequence. Inversely, if the system is designed to have one degree of freedom and the hinges get out of sync with each other, the deployable can jam as well.

**Telescopic.** Telescopic systems actuate linearly, with elements which slide linearly relative to each other, like a spyglass. It can consist of multiple elements which telescope, or can just involve linear sliding motion, like a spring exiting a canister. Stabilization is usually provided by overlapping adjacent tube segments. Examples include the magnetometer boom on QuakeSat[27], the way AggieSat1, AggieSat2[28] and RAFT[29] deployed from each other, revealing telescoping antennas, the secondary reflector on the deployable petal telescope[30].

Advantages: Deploying linearly in one direction makes it easier to build and actuation more straight forward. The design naturally provides more structural strength at the root of a telescoping boom compared to foldable or coilable booms.

Disadvantages: Motion only occurs in one direction for a telescope, limiting where the final deployable is located at. Also, to reach any length, the telescope needs to be made up of numerous segments. Tight tolerances must be met for precise location of the tip. More overlap is needed if tubes are thin thus increasing the non-structural mass and number of tube segments needed. Play in the latches between adjacent deployed tubes is a common problem[31].

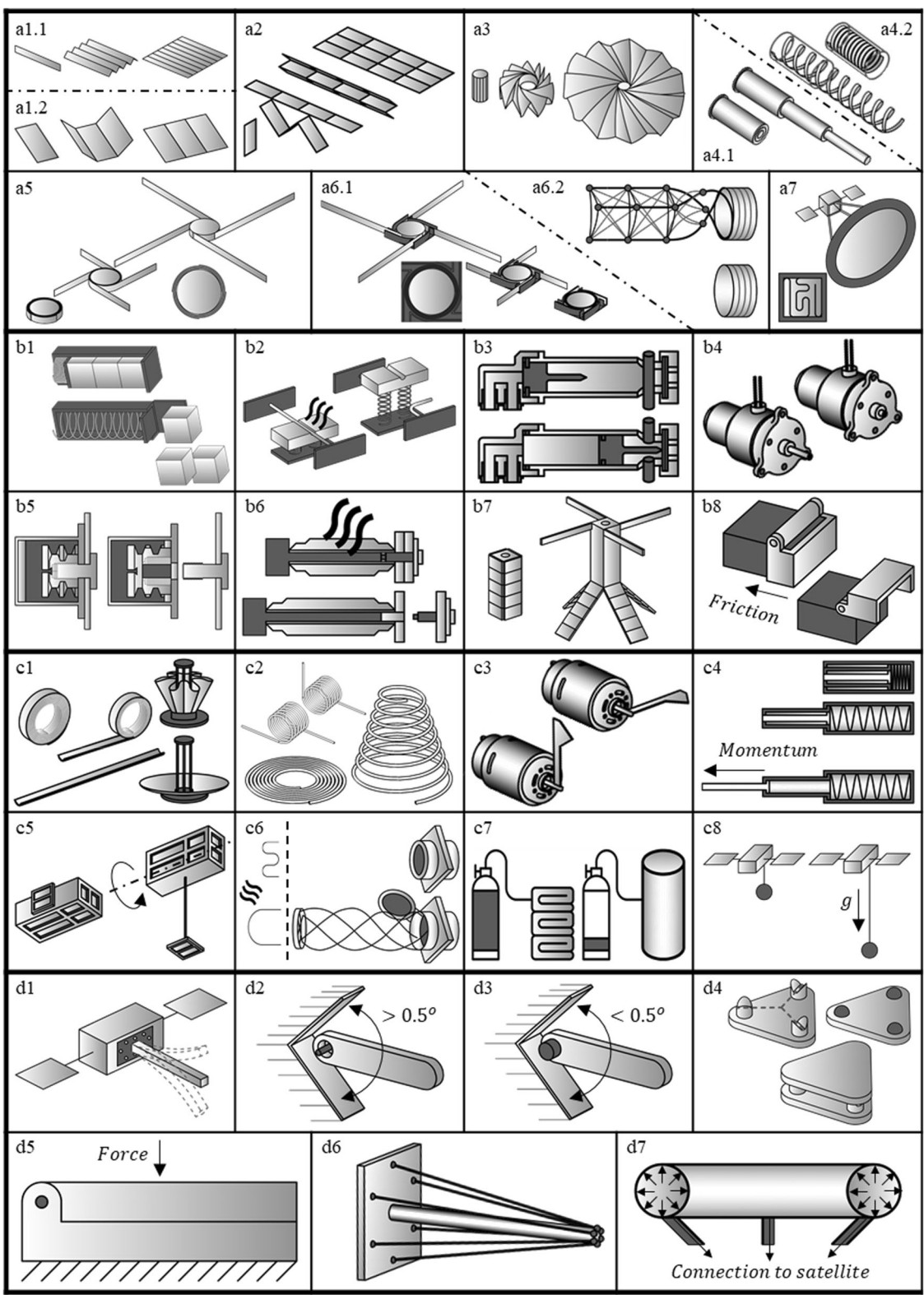

Failure Mechanisms: Friction from linear sliding or presence of foreign contaminants can lead to jamming. This is made worse if there is not appropriate overlap when sliding[32].

## Spooled

Spooled stowing methods wrap the device around a mandrel or some other structure, to keep it in a compact volume. Common spooled systems include tethers[29,33–35] and simple dipole antennas[11,36,37]. Deployable space structures that used spooling as the primary packaging architecture include the Caltech Space Solar Power Project[23], and IKAROS[38].

Advantages: Spooled systems provide an extremely compact volume to deploy a long length. When the entire deployable structure is spooled with sufficient pretension, the resulting higher stiffness offers support against

**Fig. 1 | Each approach for stowing, restraining, actuating and locating is illustrated to provide greater understanding for the text.** (a1.1) illustrates a parallel z-fold while (a1.2) illustrates a parallel tri-fold configuration. (a2) Shows both perpendicular and parallel folds and (a3) demonstrates a more complex origami fold configuration. (a4.1) illustrates a typical telescoping system while (a4.2) shows how a spring telescopes inside a canister. (a5) Illustrates a spooled system where the deployable is wrapped around a hub while (a6) illustrates a coiled system where the deployable pushes against the outer perimeter for a (a6.1) tape system and (a6.2) coilable boom. (a7) Shows the stuffed stowage configuration. (b1) illustrates a launch that can also restrain deployables. (b2) shows a burn wire mechanism that cuts a wire to enable deployment. (b3) Illustrates a pyro-cutter that could cut a bolt or cable on the right side. (b4) shows a pin puller which uses an expanding actuator to pull in a pin. (b6) illustrates a release nut that splits apart allowing the screw to move while (b6) shows a bolt breaker, that expands to stretch a bolt to its breaking point. (b7) illustrates that one deployable can sequence other deployables and (b8) illustrates

that friction or a interference fit can restrain a deployable. (c1) shows how strain energy from the deployable surface can be used to deploy the system while (c2) shows different styles of springs that can be used. (c3) indicates that motors are used for deployables. (c4) shows momentum as an approach (generally not recommended). (c5) shows that motion of the satellite can cause the deployment. (c6) illustrates that shape memory alloy wire can directly actuate the deployable after being heated. (c7) illustrates that deployables can be inflated by filling a volume with gas while (c8) shows gravity gradient can be used to deploy. (d1) illustrates a rigid mount to the spacecraft. (d2) Shows a loose and (d3) shows a precision hinge pins respectively with accuracies of worse than 0.5 degrees and better than 0.5 degrees. (d4) illustrates kinematic coupling that have exactly six points of contact while (d5) shows the system preloaded against a surface that could be with a hinge, or some other approach. (d6) illustrates soft goods in tension while (d7) shows an inflated system.

launch vibrations[39]. Moreover, the interlayer friction can provide damping to the system during launch vibrations[40].

Disadvantages: Extra mass and volume are required for the spool. Secondly, some spooled systems require tension to keep the system on the spool, and to prevent it from becoming a coiled system. This is often implemented with launch lock devices. Large interlayer slips due to insufficient pretension can also damage the functional elements such as solar cells on a spooled membrane. Adding restraints to avoid slipping and bird caging runs the risk of introducing new single points of failure for deployment.

Failure Mechanisms: If deployed too quickly or tension is not maintained on the spool, the system can bird cage or blossom[41], where the deployable elements expand off the spool.

### Coiled

Coiled systems are stowed when a coil is wrapped inside a compact perimeter. It differs from spooled systems, because instead of being tightly wrapped around a mandrel, the system is pushed out against a perimeter. Common examples include deploying tape springs like those on KaTENna[42], dipoles[43], solar arrays[22], and the stacer on CINEMA[44].

Advantages: The key advantage of a coiled system is that it is designed to exist pushing up against an outer perimeter of the device, so unlike a spooled system the deployer is naturally designed to deal with bird caging, and thus has robustness designed in.

Disadvantages: Having a mechanism around the perimeter of the coil, to manage the deployed state often is more complex and requires more mass than a spool.

Failure Mechanisms: There is the potential for friction related jamming which may hold the coil in place and prevent it from deploying. An example of this is discussed in a 6-meter dipole design[43].

### Stuffed

Stuffed systems are stowed in a way that is not repeatable each time. This is often used for large surfaces, for example in inflatables[45,46], where they are stuffed into a small, defined volume in no specific manner.

Advantages: Each time a stuffed system is stowed, it folds in different locations, avoiding consistent crease lines that occur in origami designs. This can avoid the need to design for fatigue.

Disadvantages: The deployment of a stuffed system is not deterministic. This means it can be harder to ensure tangling does not occur.

Failure Mechanisms: Tangling during deployment is a key failure mechanism of stuffed systems. For example, on the STS-77 inflatable antenna experiment[47], the antenna almost did not fully unfold as it inflated.

## Restrain approaches

Restraints keep the deployable from moving during launch and then allow deployment to initiate on command. In addition to the references here, we would like to draw the readers' attention to two key review papers on non-pyrotechnic restraint approaches[48,49].

### Host vehicle restraint

This restraint is released when the spacecraft separates from the larger host system. Examples include CubeSats that separate on deployment revealing features[28,50,51], and satellites that have an antenna pop out immediately once released[52] from a CubeSat deployer.

Advantages: A restraint external to the spacecraft simplifies the spacecraft design. This minimizes mass and spacecraft commands complexity. Further, deployment close to the host vehicle may allow for the host to record the deployment, providing external images.

Disadvantages: A negotiated interface between host vehicle and spacecraft is required. Further, deployment dynamics must be analyzed to ensure no unintended interactions between spacecraft and host. Many CubeSat hosts no longer allow this type of restraint due to risk to the host.

Failure Mechanisms: Failure can occur if the interface between host and spacecraft is not executed properly. For example, the deployable may push against the launch vehicle, preventing release from the canister.

### Burn wires

A hot metal wire is used to cut through a synthetic cable holding down the deployable. There are several sub-types of burn wires including actuated cable, where a polymer cable is pulled in tension across a hot element, actuated element, where the hot metal cutting element is actuated through the cable, and finally static, where the burn wire and cable are aligned closed to each other and held static[53]. Very commonly used materials are nichrome wires wound around severable nylon or vectran cables[53,54]. This method can also be combined with other restraint mechanisms[55]. Often, nichrome wires are replaced by resistors to avoid an exposed wire and the risk of oxidization during the testing phase[54].

Advantages: Burn wires are ultra-lightweight and small in form factor. They can be very cost effective and simple to implement and can be built in-house.

Disadvantages: Finding a way to keep the cable loaded when tying it into position can be challenging. Burn wire mechanisms use soft goods which come with their own workmanship and design concerns. Polymer cables may release smoke while being cut.

Failure Mechanisms: Burn wires can be heavily workmanship dependent and, given their small size, mistakes on installation can be missed. Failure may also occur if power requirements are not understood for the space environment (versus testing in an air environment).

### Pyro-cutters

When a deployable is restrained with a metal wire, metal cable, or thick rope, heat will not cut the cable. Instead, a more energetic cutting method is required. A pyro-cutter places an explosive charge behind a cutting blade. When fired, the charge accelerates the blade though a tube at the wire. A pyro-cutter was used to allow the ADEO drag sail to deploy[56].

Advantages: A high amount of force is generated, that can cut metal wires, and even bolts or nuts in some cases. This can release large amounts of holding force. This technique offers rapid motion and functional reliability.

Disadvantages: Pyro-cutters create a large shock load when firing. Further, safety precautions for the pyro-cutter must be negotiated with the launch provider, which can be challenging with a secondary payload or on shared launches.

Failure Mechanisms: Pyro-cutters are very reliable when procured from a heritage source. The main concern is that the shock load can cause surrounding mechanisms and structures to fail.

### Pin puller or pusher

This restraint actuates a pin inward or outwards, often powered by a Shape Memory Alloy (SMA)[48,57,58], electromagnet[59] or paraffin wax[48], triggering a release. The magnetometer booms on the CINEMA 1-5 CubeSats were released by a pin puller mechanism[44,60].

Advantages: Pin pullers and pushers have high flight heritage, are commercially available, and detailed datasheets are available. Mass is relatively low, mechanisms are resettable, and are easily tested prior to flight.

Disadvantages: For SMA pin puller, power required for deployment is temperature dependent. Actuation can be slow for this type of restraint.

Failure Mechanisms: Because pin pullers slide, if not properly sized friction from high normal forces can keep them from retracting. There have been instances of mechanism binding caused by thermal deformation. Further, pin pullers are difficult to make redundant[6].

### Release nuts

This restraint, often powered by a SMA, releases a nut allowing for a deployment. A release nut is held down by features in a receptor (for example the MicroLatch[61] and release nuts[22,62]). Similar to release nuts, another hold down mechanism leverages the mismatch in coefficients of thermal expansion of the materials to release[63].

Advantages: Release nuts have high reliability and flight heritage. Because the restraint is inherently a bolted joint, installation and testing is often done with common tools.

Disadvantages: The release forces may be related to bolt preload, requiring it to be measured in-situ.

Failure Mechanisms: For rotational deployments, release nuts may create a bolt race where the bolt must be clear of a joint prior to rotation or else the joint will bind. This may require a spring to move the bolt out of the release area.

### Bolt breaker

This restraint, often powered by an SMA, breaks a bolt at a weakened cross section, to release the device. These restraints are less common on small spacecraft but were used on DICE Missions[64]. Common off the shelf products include the Frangibolt[48].

Advantages: Advantages are similar to release nuts but higher preloads can be achieved.

Disadvantages: Breaking the bolt causes debris which must be caught, and the catch can make the restraint too large for some CubeSats. The process of breaking the bolt creates a high shock load.

Failure Mechanisms: Because the bolt is intended to be weakened, it is possible to over torque during installation and begin the release unintentionally.

### Another deployable

Deployable structures can be stacked on top of each other, such that releasing one will release subsequent deployables. For example, the antennas on QuakeSat were released when the solar panels opened[36] and the University of Tokyo's XI-IV CubeSat used one large antenna to hold down two smaller antennas[37].

Advantages: Limiting the number of releases reduces power draw, commands required for deployment, mass and undesirable release loads going into the system upon release.

Disadvantages: Combining numerous deployments into a single actuation requires great care to consider deployment dynamics, forces and synchronization. In an anomaly situation it may be difficult to trouble shoot deployment issues when numerous degrees of freedom are tied to a single release.

Failure Mechanisms: If not synchronized, the deployables could jam against each other. If the first deployable fails during deployment, it can prevent subsequent deployments.

### Latch/Tight Fit

Latch/Tight Fit occurs when enough soft preload is applied to a deployable or cover, to restrain it during launch. After launch, the activation of the deployment actuator pushes the restraint open. The inflatable drag sail on AeroCube 3 is an example[65].

Advantages: These systems can be very simple and lightweight. They can often be built to suit the application and incorporated into system structures.

Disadvantages: The actuation method must be strong enough to overpower the restraint, which often is much more force than what is required for deployment.

Failure Mechanisms: This restraint is held in place by friction, which can result in premature release if friction is too low, or prevent deployment if friction is too high.

## Actuate approaches

These are approaches for which energy is transferred to move the deployable.

### Strain energy

Strain Energy is the simplest actuation approach, where the device is deployed by its own strain energy. For example, as often seen in simple dipole antennas, a tape measure is simply bent over and springs back to its shape under its own energy[11,36,37,66]. Other examples include deployables that utilize high-strain composites for dipole antennas[21], flexible solar arrays structures[23], helical antennas[67], and wrap-rib antennas[68–70].

Advantages: Strain energy can easily be released using numerous types of restraints. This actuation method requires no power draw. If the strain energy also deploys out of the plane of deployment (like many rollable booms), the deployed state can be stiffened.

Disadvantages: Strain energy hardware is often subject to large displacements and may require specialized analysis for creep and low-cycle fatigue.

Failure Mechanisms: Strain energy deployments are often fast and can be uncontrolled. Since multiple stable configurations are possible, structure may not reach the intended fully deployed configuration if it is not properly guided. The design must ensure that the deployment does not impact sensitive systems of the spacecraft or is subject to damaging/tangling itself.

### Springs

Springs store strain energy, and then move when released. They are differentiated from the strain energy actuation approach as springs are a separate component of the deployable. Different types of springs include compression[17,71,72], conical compression, extension, torsion[17], tape, and constant force springs[71]. Torsion springs are often used within hinges for solar panels[5,9–11].

Advantages: Springs have high amounts of flight heritage, are highly customizable, and commercially available.

Disadvantages: Spring force changes with displacement, changing force must be accommodated. This can be mostly reduced with constant force springs, but not entirely eliminated.

Failure Mechanisms: Springs are subject to fatigue, and consideration should be given to how the spring operates if it were to break. This is why compression springs are preferred over extension springs. Torsion springs are subject to frictional losses. Further, springs may cause binding in mechanisms if not implemented with proper tolerancing. CAD models often do not consider both deformed and relaxed states, which may lead to issues upon assembly.

### Motors

Motors are another common actuation method. However, as motors require a power source and electronics to drive them, if a motorized

actuation can be replaced with a spring, it often is. Direct current motors for deployment are broken into brushed and brushless motors. For deployments, motors are often used to spool booms or actuate a latch[73].

Advantages: Motor actuation offers unparalleled ability to control and track deployments. If a deployment anomaly occurs, motors can be commanded in both directions to attempt to fix the issue.

Disadvantages: Motors often require controller electronics and higher power draws. Motors must often be paired with other mechanisms to maintain stiffness once deployed. It is difficult to make motors redundant on small spacecraft.

Failure Mechanisms: Motor electronics must be tested for EMI compatibility with a system. Further, motor gear boxes must be specially designed for the space environment (e.g., thermal, lubrication). Finally, caution should be taken in sizing the motors and attached gearboxes, to ensure that if a stall occurs, the motor will not strip the gears.

## Momentum

Sometimes momentum from the initial restraint release is used to also actuate the deployment. This has been often observed in tethers with mixed success[28,50].

Advantages: Combining restraint release and deployment actuation results in a single command and low power draw.

Disadvantages: Combining restraint and deployment limits options available during anomalous situations. There is only one opportunity for the deployment to occur.

Failure Mechanisms: If the restraint method fails, the deployment actuation also fails. Further if deployment stops at any time before reaching the final state, the deployment will not restart if the stopping force is removed, unlike other methods. This is why standards do not allow for the use of momentum to deploy[32].

## Satellite dynamics

Motion of the satellite can be used to actuate a deployable, for example, a spinning satellite can tension a membrane or cables. Examples of space missions include the deployment of the 30 m tether on the CUTE 1.7 CubeSat[74] and IKAROS[38].

Advantages: This approach utilizes the existing spacecraft system to actuate the deployment, without the need for additional components.

Disadvantages: Spacecraft motion relative to deployable performance must be well characterized. This may be difficult to test on the ground and deployment forces are limited.

Failure Mechanisms: The deployable can interfere with the ability of the satellite to control itself.

## Shape memory alloys

While Shape Memory Alloys (SMA) are used primarily in restraint devices, they can also be used to actuate a deployment, as observed in helical antennas[75], solar arrays[76], and solar sails[76]. Alternatively, shape memory alloys can be employed for the hold down and release[77,78], as was used on the ISARA spacecraft[79]. For a collection of best practices in SMA-actuator design, see ref. 80.

Advantages: SMA actuators operate slowly, unlike the dynamic spring and strain energy deployments. They offer a reduction in the weight, volume, and overall complexity of the system, and can operate taking advantage of the ambient thermal environment[80].

Disadvantages: SMA are inherently temperature correlated. Some SMA exhibit non-linear characteristics that need to be fully understood.

Failure Mechanisms: Hardware can sometimes be fragile and fail during testing without obvious signs.

## Inflation

Inflation can be used to actuate elements. There are multiple ways of storing compressed gas, which can then be released into a piston or an enclosed volume, causing it to expand. AeroCube-3 used an inflatable balloon as a de-orbit device but failed to inflate on orbit[65]. Technologies have been explored for inflatable antennas and a planned demonstration on CatSat[45,81].

Advantages: Inflation based actuation results in high efficiency packing envelopes that can be made in customized sizes to match available resources.

Disadvantages: Inflation can be an effective actuation method but can be hard to test in a flight-like setting, due to pressure/temperature differences in space.

Failure Mechanisms: Leaking and fatigue may occur because of workmanship errors with the use of soft goods.

## Gravity gradient

Gravity gradient, or the difference in gravitational forces applied to two objects at a distance from each other, can be used to deploy a system very slowly. This is often observed in tether systems[35].

Advantages: Gravity gradients inherently leverage physics and can result in very efficient packing ratios. Actuation is smooth and slow.

Disadvantages: Gravity gradient deployments are often reliant on the attitude control system of a spacecraft and may require coordination between subsystems. Further, gravity gradients often require large displacements (1 m + ) to be effective.

Failure Mechanisms: Failure of the attitude control system or saturation, may result in the gravity gradient failing.

## Locate approaches

Location is how the deployable is maintained in its final deployed state.

## Rigidly attached to structure

Rigidly Attached Structure generally applies to strain energy deployments, which are directly fixed to the structure. This is the most common way dipole antennas are constrained to a small satellite. An example includes the deployable mast in AAReST[5].

Advantages: This approach is simple and repeatable.

Disadvantages: A single rigid connection point often results in a long, cantilevered beam which can have a low first mode.

Failure Mechanisms: When pushed to the extreme, long cantilevered beams may lead to the satellite attitude control system (ACS) losing control due to resonances or flutter. This was observed on a 3-meter dipole design[43].

## Loose hinge

A loose hinge is defined where the deployment accuracy will be worse than 0.5 degrees. This is how most CubeSat solar panels[10,11] are located.

Advantages: A loose hinge is reliable, low cost and easy to design.

Disadvantages: The deployment is not highly accurate.

Failure Mechanisms: If the barrel of the hinge, which surrounds the hinge pin, is too thin, or its connection to the leaf is too thin, it can yield with repeated impacts at the end of deployments, reducing deployment accuracy.

## Precise hinge

Precise hinges provide deployment accuracy of better than 0.5 degrees. This is accomplished by a high tolerance hinge pin, hinge pin hole, and hard stop distanced from the hinge. These are commonly used for precision applications like reflectarray antennas[17,82] and telescope mirrors[30].

Advantages: Precision hinges deploy with greater repeatability, which is critical for systems like antennas. Adjustability is sometimes built into the hinge hard stop, to enable tuning of the angle after final assembly, relaxing manufacturing requirements[82].

Disadvantages: To get the precise deployment, extra design effort, mass, and structural depth is required.

Failure Mechanisms: In addition to loose hinge pin failure mechanisms, the adjustable hard stop can shift over repeated deployments or launch loads, causing a non-repeatable deployment. Further, hinges may have redundant constraints which make a non-repeatable deployment angle, depending on what is in contact with each other.

## Kinematic mounts

Kinematic Mounts can be used to control the position of the system with a high degree of precision. A kinematic mount ensures exact constraint of all 6 degrees of freedom[83]. This means there is consistently one, and only one location the system will end up in. The kinematic contacts are often arranged in a Maxwell (three V-groves) or Kelvin (tetrahedron, V-grove and plane) configuration[83].

Advantages: Kinematic mounts result in extremely precise locations, down to micron level repeatability, with most of the errors being due to material deformation.

Disadvantages: Kinematic mounts often add an additional separate system not used in deployment, increasing mass. Design for kinematic mounts requires more experience.

Failure Mechanisms: An adequate nesting force is required to hold the deployable in the final position. If the force is not adequate or in the wrong direction, there may be less than the required 6 points of contact. The 6 points of contact must not be redundant with each other. Redundant points of contact can result in both an over constrained and under constrained system, where final position is not determinate[84].

## Preloaded against a surface

This method is a more general case of the hinges and kinematic mounts previously discussed where a preload holds the deployed system against a static surface. This category is a catch all for preloading positioning methods that are less common; for example, the floating core concept for deployable double omega booms by DLR[85].

Advantages: Preloading against a surface tends to increase the stiffness of the deployed structure over some other methods. It also can be uniquely customized for various applications.

Disadvantages: Location mechanisms to provide the preload tend to be more complex than rigidly attaching to structure.

Failure Mechanisms: Often these designs are implemented without considering exact constraint, which means that for high accuracy system, over constraint can be an issue. Further, over constrain can result in high loads during or after deployment due to tolerances or thermal effects.

## Softgood(s) in tension

Locating cables or a fabric sleeve can be used to hold a deployable in a specific position. The deployable has a nesting force tensioning the soft good. Examples include cables to precisely position an antenna feed[82], a UHF antenna over a ground plane[17], and sleeves used to add stiffness to SHEARLESS lenticular booms[86].

Advantages: One key advantage of using softgoods is that they are highly compact and can control position over a long deployment distance.

Disadvantages: A key challenge is cable or softgood management. Sometimes these are easier, like when they surround a SHEARLESS boom. But in other cases, systems must be designed to manage coils and prevent them from tangling, like on the space solar power demonstration (SSPD1) mission[23,87]. Softgoods are subject to higher design safety factors and specific workmanship concerns.

Failure Mechanisms: Tangling of the softgoods can prevent deployment. After deployment creep of the softgoods can change deployed position over time.

## Inflation

Inflation is a special case of a soft good in tension, where the deployed state is constrained by the forces due to pressure. Instead of an actuator providing a preload in one direction, pressure expands against the entire surface. This is commonly observed in deployable antennas[45,46] and drag devices[65].

Advantages: Inflation can be used to control the position of a large shape.

Disadvantages: Inflation tends to push all shapes towards spherical or cylindrical geometries[88]. If a non-rounded shape is desired, it is hard to achieve and requires additional elements in the design. Tension on the inflated surface can also result in wrinkles on the surface.

Failure Mechanisms: An inflatable surface will fail if pressure is lost. To mitigate this, designs have been investigated where inflation achieves an initial shape, and then curing (UV, thermal or other chemical process in space)[89] or yielding of a metal shell[90] is used to maintain the final shape.

## Relationships between approaches

Table 1 illustrates which approaches for each stage are respectively observed together for small satellites, indicating usually observed together (++) and often observed together (+). Table 1 should be read as the first vertical column is usually/often observed with the first row. Note that the relationships vary. For example, host vehicle restraints are uncommon, but when they occur, they are observed with parallel and orthogonal folds. This means launch vehicle restraints have a relationship with parallel/orthogonal folds, but parallel/orthogonal folds are not related to launch vehicle restraint. The numbers in the Table 1 align with the Fig. 1.

## Key deployable design practices observed

To guide implementation of deployable approaches, design practices follow. The design practices are divided into two subcategories, general and implementation specific. General apply to nearly all the approaches, whereas implementation specific apply to a subset of the approaches. Each design principle is illustrated in Fig. 2.

## General design practices
### Reduce the number of deployments

The first practice is to reduce the number of deployments, or actuations required to deploy a structure. The most basic implementation is to avoid deployables. Before designing a deployable structure, think creatively through the spacecraft configuration to ensure a deployable is really required. If a fixed solution is not an option, keep the deployment in as few planes as possible. The fewer the planes, the easier it is to offload and test the deployable. Also reduce the number of actuators, first in parallel, and then in series with each other. This increases deployment reliability.

### Prototype early and often

While good analysis techniques are important for developing deployables and can decrease costs, many deployables are complex and deployments are non-linear, making accurate analysis challenging. It is critical, especially for small satellites where budgets are limited, to quickly design, build, and test, working through as many iterations as possible to improve the design. Initially, prototyping should start at a low fidelity, and gradually increase as the design becomes more refined[91].

### Test as you fly early

During development, testing as close to how the deployable will fly is important, as soon as practical. (Caveat: it is important to know the fidelity of early prototypes, and not subject them to tests that are too extreme (e.g., it is impractical to test the first prototype in thermal vacuum right away) and not slow down the prototyping iteration speed to develop a refined prototype.) Tests should get closer and closer to flight conditions (e.g., gravity offloading, thermal), to build a set of data that is relevant to how the deployable will perform on orbit. Electronics tend to perform poorly in thermal vacuum at hot temperatures, lubrications in mechanisms tend to be influenced by cold temperatures (cold ambient pressure testing is often allowable), and stowed structures tend to be driven by vibration and shock loads. When it is not practical to test the exact condition that will be observed on orbit, a more extreme condition can be tested on the ground (for example, thermal ambient pressure at a higher temperature than thermal vacuum for hot conditions). Detailed guidelines for testing specific for deployables have been written[92]. Finally, it is important to be collecting the same data in the same way the spacecraft will be collecting it (in addition to any ground telemetry), as this will provide a valuable data set that can be used to investigate any on-orbit anomalies[91].

**Table 1 | Relationships between approaches and approaches of other stages**

The approach in the column is (usually ++, often +) observed with the approach in the row.

| | Stow Approaches | | | | | | | Restrain Approaches | | | | | | | |
|---|---|---|---|---|---|---|---|---|---|---|---|---|---|---|---|
| | a1 Parallel Fold(s) | a2 Orthogonal/Parallel Folds | a3 Origami/Kirigami Folds | a4 Telescopic | a5 Spooled | a6 Coiled | a7 Stuffed | b1 Host Vehicle Restraint | b2 Burn Wires | b3 Pyro-Cutter | b4 Pin Puller or Pusher | b5 Release Nuts | b6 Bolt Breaker | b7 Another Deployable | b7 Latch/Tight Fit |
| **Stow Approaches** a1 Parallel Fold(s) | | | | | | | | | ++ | + | | + | + | | |
| a2 Orthogonal/Parallel Folds | | | | | | | | | ++ | + | | + | + | + | |
| a3 Origami/Kirigami Folds | | | | | | | | | + | + | | + | | | + |
| a4 Telescopic | | | | | | | | | | + | ++ | + | + | | |
| a5 Spooled | | | | | | | | | ++ | | + | | | + | |
| a6 Coiled | | | | | | | | | ++ | | + | | | + | |
| a7 Stuffed | | | | | | | | | | | + | | | | ++ |
| **Restrain Approaches** b1 Host Vehicle Restraint | + | + | | + | | + | | | | | | | | | |
| b2 Burn Wires | ++ | + | | | + | + | | | | | | | | | |
| b3 Pyro-Cutters | + | + | + | + | | | | | | | | | | | |
| b4 Pin Puller or Pusher | | | + | + | + | + | + | | | | | | | | |
| b5 Separation Nuts | ++ | + | + | + | | | | | | | | | | | |
| b6 Bolt Breaker | ++ | + | + | | | | | | | | | | | | |
| b7 Another Deployable | + | + | + | | + | ++ | | | | | | | | | |
| b8 Latch/Tight Fit | | | + | | ++ | + | ++ | | | | | | | | |
| **Actuate Approaches** c1 Strain Energy | | + | + | | ++ | + | | | ++ | | | | | | |
| c2 Springs | ++ | + | + | | | | | | ++ | | + | | + | + | |
| c3 Motors | | + | + | + | + | ++ | | | | | + | | + | + | |
| c4 Momentum | | | | ++ | + | | | | ++ | | + | + | + | + | |
| c5 Satellite Dynamics | | | | ++ | + | | | | ++ | | + | + | + | + | |
| c6 Shape Memory Alloy | ++ | + | + | | | + | | | ++ | | + | + | | | + |
| c7 Inflation | | + | + | + | | | ++ | | | | | | | | |
| c8 Gravity Gradient | | | | | ++ | | | + | + | | | | | | |
| **Locate Approaches** d1 Rigidly Attached | + | + | | | ++ | + | | | ++ | | | + | + | + | + |
| d2 Loose Hinge Pins | ++ | + | + | | | | | | ++ | | | + | + | + | + |

## Table 1 (continued) | Relationships between approaches and approaches of other stages

**The approach in the column is (usually ++, often +) observed with the approach in the row.**

| | Stow Approaches | | | | | | | Restrain Approaches | | | | | | | |
| --- | --- | --- | --- | --- | --- | --- | --- | --- | --- | --- | --- | --- | --- | --- | --- |
| | a1 Parallel Fold(s) | a2 Orthogonal/Parallel Folds | a3 Origami/Kirigami Folds | a4 Telescopic | a5 Spooled | a6 Coiled | a7 Stuffed | b1 Host Vehicle Restraint | b2 Burn Wires | b3 Pyro-Cutter | b4 Pin Puller or Pusher | b5 Release Nuts | b6 Bolt Breaker | b7 Another Deployable | b7 Latch/Tight Fit |
| d3 Precise Hinge Pins | ++ | + | + | | | | | | ++ | + | + | + | + | | |
| d4 Kinematic Mounts | + | ++ | + | + | | | | | | | + | + | + | | |
| d5 Preloaded Surface | | | | ++ | + | + | | | + | | + | + | + | | |
| d6 Tensioned Softgood | | | + | | + | | ++ | | + | | + | + | + | | + |
| d7 Inflation | | | + | + | | | ++ | | | | | | | | ++ |

**The approach in the column is (usually ++, often +) observed with the approach in the row.**

| | | Actuate Approaches | | | | | | | | Locate Approaches | | | | | | |
| --- | --- | --- | --- | --- | --- | --- | --- | --- | --- | --- | --- | --- | --- | --- | --- | --- |
| | | c1 Strain Energy | c2 Springs | c3 Motors | c4 Momentum | c5 Satellite Dynamics | c6 Shape Memory Alloys | c7 Inflation | c8 Gravity Gradient | d1 Rigidly Attached | d2 Loose Hinge Pins | d3 Precise Hinge Pins | d4 Kinematic Mounts | d5 Preloaded Surface | d6 Tensioned Softgood | d7 Inflation |
| Stow Approaches | a1 Parallel Fold(s) | + | ++ | | | | | | | | ++ | + | | | | |
| | a2 Orthogonal/Parallel Folds | ++ | ++ | | | | | | | | ++ | + | + | | | |
| | a3 Origami/Kirigami Folds | ++ | | + | | | | | | + | ++ | | | | | |
| | a4 Telescopic | + | ++ | + | | | | | | ++ | | | | + | + | |
| | a5 Spooled | ++ | + | + | | + | | | | ++ | | | | | | |
| | a6 Coiled | ++ | + | + | | | | ++ | | ++ | | | | + | | |
| | a7 Stuffed | | | | | | | | + | + | | | | | + | |
| Restrain Approaches | b1 Host Vehicle Restraint | ++ | + | | | + | | | | ++ | + | | | | + | |
| | b2 Burn Wires | ++ | + | | | | | | | ++ | + | | | | + | |
| | b3 Pyro-Cutters | + | + | | | | + | + | | ++ | + | + | | + | | |
| | b4 Pin Puller or Pusher | + | ++ | + | | | | | | + | ++ | + | + | + | + | |
| | b5 Separation Nuts | ++ | + | | | | | | | | ++ | + | + | + | | |
| | b6 Bolt Breaker | ++ | + | | | | | | | | ++ | + | + | + | | |
| | b7 Another Deployable | ++ | + | | | | | | | ++ | + | + | + | + | + | |

**Table 1 (continued) | Relationships between approaches and approaches of other stages**

| | Actuate Approaches | | | | | | | | Locate Approaches | | | | | | |
|---|---|---|---|---|---|---|---|---|---|---|---|---|---|---|---|
| The approach in the column is (usually ++, often +) observed with the approach in the row. | c1 Strain Energy | c2 Springs | c3 Motors | c4 Momentum | c5 Satellite Dynamics | c6 Shape Memory Alloys | c7 Inflation | c8 Gravity Gradient | d1 Rigidly Attached | d2 Loose Hinge Pins | d3 Precise Hinge Pins | d4 Kinematic Mounts | d5 Preloaded Surface | d6 Tensioned Softgood | d7 Inflation |
| b8 Latch/Tight Fit | | | | | | | ++ | + | | | | | | + | ++ |
| **Actuate Approaches** c1 Strain Energy | | | | | | | | | ++ | | | | | | |
| c2 Springs | | | | | | | | | | ++ | + | | + | + | |
| c3 Motors | | | | | | | | | | + | + | + | ++ | + | |
| c4 Momentum | | | | | | | | | ++ | | | | + | | |
| c5 Satellite Dynamics | | | | | | | | | ++ | | | | + | + | |
| c6 Shape Memory Alloy | | | | | | | | | ++ | | | | | | |
| c7 Inflation | | | | | | | | | | | | | | | ++ |
| c8 Gravity Gradient | | | | | | | | | + | | | | | ++ | |
| **Locate Approaches** d1 Rigidly Attached | ++ | | | | | | | | | | | | | | |
| d2 Loose Hinge Pins | | ++ | + | | | | | | | | | | | | |
| d3 Precise Hinge Pins | | ++ | + | | | | | | | | | | | | |
| d4 Kinematic Mounts | | | + | | | | | | | | | | | | |
| d5 Preloaded Surface | | ++ | + | | | | | | | | | | | | |
| d6 Tensioned Softgood | | + | + | | + | | ++ | | | | | | | | |
| d7 Inflation | | | | | | | ++ | | | | | | | | |

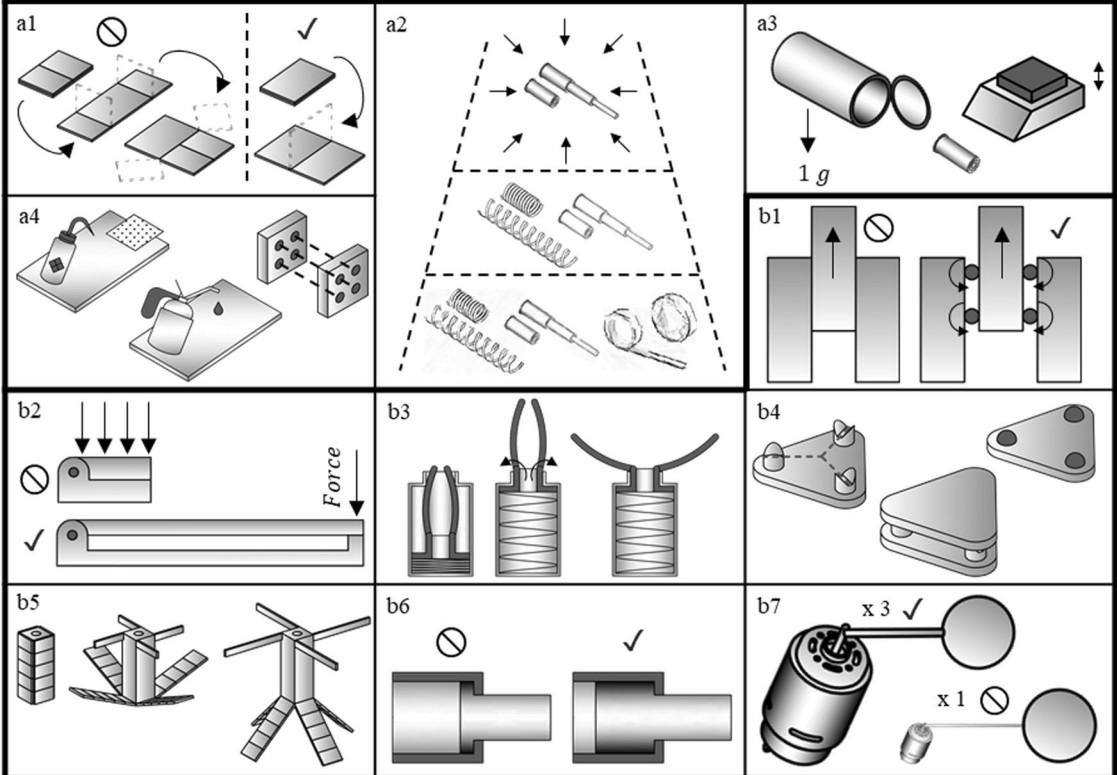

**Fig. 2 | An illustrated guide to the design practices, starting with general architecture agnostic practices, followed by best practices for specific approaches.** (a1) is represents reducing the number of deployment panels, (a2) represents prototyping early and often with many solutions, and then to refine the concepts until the right solution is reached, (a3) illustrates testing as the system flies in thermal vacuum, vibration testing, and notes gravity effects, (a4) illustrated that care must be taken with interfaces, from surface preparation to reducing the number of locations where interfaces interact with each other. (b1) Illustrates that rotary motion is preferred over sliding, (b2) shows to use structural depth in combination with rotation for precise placement and (b3) shows the use of self-help, where geometry is used to redirect forces. (b4) Illustrates an exactly constrained system, with six points of contact and (b5) illustrates deployable self-sequencing. (b6) Shows that if sliding is required, large rations of the length/diameter should be utilized and (b7) illustrates that significant force/torque margin should be used.

## Give extra time to interfaces

Many issues occur at the interfaces of hardware and systems. Traditionally interfaces are thought of as where a deliverable from one party meets the deliverable from another party, for example, payload to spacecraft bus or spacecraft bus to launch vehicle. While these interfaces often bring up problems (even when both parties think there is clear communication, only to realize it was unclear to the other party), and deserve extra attention, it is important to note any interface of motion or material change in the deployable. For example, sliding and rolling interfaces need to be lubricated with dry or solid film lubricant. It is worth highlighting that for commercial parts, standard lubricants may need to be replaced with vacuum rated lubricant, or the lubricant may evaporate before deployment. Further, if similar materials contact each other, (for example a stainless-steel bolt in a stainless-steel insert are frequent problem areas) the metals can gall or cold weld.

An increasing number of deployables use composite surfaces, which are commonly bonded at the interfaces. When bonding, proper abrasion and cleanliness approaches are paramount to ensure adequate strength. Further, the geometry should be configured to avoid peeling, for which bonded joints have low strength. Failures on some early MarCO prototype panels occurred due to lack of control in the bonding environment[2] and bonding cleanliness was a key lesson learned in the Milstar solar array development[15].

## Implementation specific design practices
### Use rotational over linear motion
Rotational motion reduces the effects of friction over sliding. Given friction can be variable in the vacuum environment and with temperature, it is desirable to design mechanisms to operate with rotary motion instead of linear sliding. The effect of friction is reduced by rotary motion through two approaches: moment arms and rolling. Rotary motion can be designed (for example in a linkage) such that friction is acting on a short moment arm, whereas the driving force is applied to a longer moment arm, further reducing the effect of friction[84]. If linear motion is unavoidable, rolling elements can be added between the two linear sliding parts, as are often observed in boom deployers[93], as pure rolling (aside from rolling resistance) has no resistive friction force.

### Use structural depth with rotation
Like using moment arm distances to increase ratios between the input force and friction, structural depth is also helpful in rotary systems to decrease loads and improve deployment accuracy. For example, in a hinge that deploys, placing a hard stop as far away as possible from the hinge pin can greatly improve deployment accuracy, and was implemented on the RainCube and OMERA antennas[82,91].

### Self help
Self-help is defined as using geometry to increase, create, or redirect forces. Creative design on a deployable's geometry can result in self-help, which can increase deployment force margins or reduce the number of actuators required. Structural depth could be viewed as a special case of self-help, as depth decreases forces required and increases structural margin. Other examples of self-help include how the RainCube antenna used its upward motion moving out of the canister, to also rotate and deploy the root ribs[71]. Another example is how DOLCE[23] adds a four-bar linkage to increase the holding force of a launch lock device.

**Table 2 | Relationships between approaches and implementation specific design practices. General design practices do not appear in the table, as they apply to nearly all approaches**

| | Symbols indicate strong (++), moderate (+), or negative (-) connection between the deployment approach and design practice. | b1 Rotational vs Linear Motion | b2 Structural Depth with Rotation | b3 Self Help | b4 Exact Constraint | b5 Self-Sequencing | b6 High Aspect Ratios when Sliding | b7 Force/Torque Margin |
|---|---|---|---|---|---|---|---|---|
| Stow Approaches | a1 Parallel Fold(s) | ++ | + | + | + | + | | + |
| | a2 Orthogonal/ Parallel Folds | ++ | + | + | + | ++ | | + |
| | a3 Origami/Kirigami Folds | + | + | ++ | ++ | ++ | | + |
| | a4 Telescopic | - | | | + | | ++ | + |
| | a5 Spooled | ++ | | | | | + | + |
| | a6 Coiled | ++ | | | | | | + |
| | a7 Stuffed | | | | | ++ | + | |
| Restrain Approaches | b1 Host Vehicle Restraint | | | | + | ++ | + | ++ |
| | b2 Burn Wires | + | | + | + | | + | + |
| | b3 Pyro-Cutters | - | | + | | | ++ | |
| | b4 Pin Puller or Pusher | - | | + | + | | ++ | + |
| | b5 Separation Nuts | | | + | + | | + | + |
| | b6 Bolt Breaker | | | + | | | | + |
| | b7 Another Deployable | + | | ++ | + | ++ | | + |
| | b8 Latch/Tight Fit | + | | + | - | + | | ++ |
| Actuate Approaches | c1 Strain Energy | | | + | | + | | + |
| | c2 Springs | ++ | + | + | | + | | ++ |
| | c3 Motors | ++ | + | + | | | | ++ |
| | c4 Momentum | | | | | + | | - |
| | c5 Satellite Dynamics | | | ++ | | | | |
| | c6 Shape Memory Alloy | + | | + | | | | + |
| | c7 Inflation | | | + | | + | | |
| | c8 Gravity Gradient | | | ++ | | | | |
| Locate Approaches | d1 Rigidly Attached | | | | + | | | |
| | d2 Loose Hinge Pins | + | ++ | | + | | | |
| | d3 Precise Hinge Pins | + | ++ | | ++ | | | |
| | d4 Kinematic Mounts | | ++ | | ++ | | | |
| | d5 Preloaded Surface | | + | | + | | | |
| | d6 Tensioned Softgood | | | | ++ | | | |
| | d7 Inflation | | | | | + | | |

## Utilize exact constraint

Utilize exact constraint where possible, where each of the six degrees of freedom are removed with non-redundant constraints[94]. This applies to both structures and locating elements. This not only results in accurate deployments, as discussed earlier, but also increases reliability and structural efficiency, reducing mass. Keeping load paths simple and non-redundant makes it less likely that differences in thermal expansion will cause structural failure.

## Self-sequencing of deployables

Sequencing, and self-sequenced designs can be helpful to reduce the number of actuators and avoid interference during deployment. This can cause deployments to happen in a specific order, like the deployable implementation on the RainCube antenna where actuation and geometry first deployed the antenna out of the canister, then deployed the root ribs, then the tip ribs, and finally the sub-reflector. Most orthogonal/parallel folds are self-sequenced, as well as origami designs. Systems which use another deployable as a launch restraint, like Prometheus[10] are also self-sequenced.

## Use high aspect ratios when sliding

When sliding is unavoidable, it is important to ensure a high aspect ratio, also referred to as length over diameter or L/D. This is minimum length of an overlapping section between the two sliding elements in the furthest deployed state vs the diameter of the deployable. Inadequate L/D has led to issues with deployables jamming during test[91]. It is generally recommended that L/D be equal to or greater than 2:1, unless it can be proven through analysis that a lower ratio is not an issue[32].

## Force/Torque margin

Best practices recommend that deployment mechanisms have a force or torque margin of three times the required deployment force when obtained via analysis, two times when flight like hardware is built and tested, and one times for a one spring out case (where it is assumed one of the springs fail)[32]. However, for small satellites, which are developed quickly and on a higher risk posture, it is recommended to use even higher torque margins. The key concern comes when the additional force or torque costs mass or results in undesirable deployment dynamics.

## Table of implementation specific design practices vs deployment approaches

To help understand what design practices are most important to apply to which deployment approaches, Table 2 has been created. This table shows if there is a strong (++), moderate (+), or negative (-) connection between the

deployment approach and design practice. Weak or not applicable as shown as blank in the table. This will help an engineer determine which design practices should be paid careful attention to when implementing a specific approach.

## Outlook: challenges and future directions

From creating this framework, studying approaches, and design principles, the authors observed three key challenges facing Small Satellite deployables today and three future directions related to each challenge.

### Deployed aperture size remains static, while instruments continue to shrink

The motivation behind this paper stems from the relentless miniaturization of instruments enabling small satellites to do real science. While instruments are continuing to shrink, enabling even smaller spacecraft, requirements for aperture size remain unaltered, primarily dictated by the physics of the measurements or ground footprints. This poses two challenges.

First, the deployable mechanisms are growing to a notable fraction of the spacecraft's size and mass. This, in turn, presents issues related to attitude control, during both deployment and post-deployment phases, potentially undermining the spacecraft's controllability. Moreover, thermally induced deformations can disrupt the control system.

Second, deployables have the potential to influence spacecraft size, much like non-deployable aperture size had in the past. As instruments get smaller, deployable apertures could drive payload mass and volume, making the deployable a key cost driver.

### Increased capabilities of small satellite instruments create stringent requirements

As instruments continue to shrink, their capabilities are expanding in complexity. This expansion imposes more stringent requirements on deployment accuracy, driven by higher electromagnetic operational frequencies or advanced measurement techniques like interferometry. The evolving sophistication of instruments, bolstered by electronics advancements, is placing increased demands on deployable mechanisms, which is a mature field. This creates a challenge that the development of mechanism technology lags behind the growing demand for performance from instruments.

### Professionalization of small satellites scope creep and cost growth

The professionalization of small satellite development is reducing the tolerance for failure in a field that traditionally accepted it due to low costs. This trend parallels the evolution of space missions from the early satellite era to the post-Apollo era. The resulting challenge is as mission scope and low cost expectations increase, the appetite for risk in small satellite deployables is diminishing, intensifying the difficulties in addressing the initial two challenges.

### New opportunity: in-space assembly, precise formation flying, and low altitude smallsats

To address the challenge of shrinking instruments and static deployables, three new opportunities arise: in-space assembly, precise formation flying, and low-altitude small satellites. These opportunities promise a larger coverage area with minimal launch volume requirements. In-space assembly and manufacturing increase the deployed area compared to stowed volume. This efficiency stems from shifting deployment mechanism complexity into robots, allowing for mechanism reusability across multiple locations. A key technology is miniaturizing robots for small satellites. Precision formation flying can provide an alternative to large deployables, or be used in combination with smaller deployables to enable synthetic aperture and interferometry. A key technology is highly accurate ACS systems and propulsion. Finally, low altitude small satellites enable smaller apertures, as they are closer to the ground. One key enabling technology for low altitude small satellites are propulsion systems to counteract increased drag.

### New opportunity: build the deployable at the material level

The field of mechanisms has seen slow development, particularly when compared to the dramatic advancements in electrical systems over the past 80 years. However, new opportunities arise through composites, additive manufacturing, and metamaterials. Composites offer not only reduced weight but also tailored material properties and high strain. Additive manufacturing allows for unique geometries and cost-efficient production based on part volume. Metamaterials can be used in RF design to eliminate the need for deployable feeds and in the mechanical design to introduce flexures into parts, incorporating mechanisms at the material scale, enabling highly miniaturized mechanisms[95].

### New opportunity: AI assisted design of deployables

To address the challenge of reduced tolerance for failure in the face of increasing expectations in the small satellite sector, there is a need to facilitate the quick adoption of robust practices among those new to the field. Machine learning and artificial intelligence tools have the potential to assist deployable space mechanism designers by suggesting novel mechanisms and identifying potential issues before testing. Topology optimization has already showcased the efficiency gains achievable through generative design tools. In the future, a software-driven approach could parameterize deployable challenges, suggesting methods and highlighting potential failure modes, even proposing entirely new, unexplored solutions. The foundation laid by this paper offers an opportunity for someone to develop such a model.

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

## Acknowledgements

The authors would like to thank Dr. Manan Arya for the insightful inputs he provided when initially identifying various approaches for each deployment stage, which occurred during discussions for writing an earlier survey paper[4]. The authors would like to thank the many authors from the references, who documented their work, and identified challenges in their Small Satellite deployables. The research was carried out at the Jet Propulsion Laboratory, California Institute of Technology, under a contract with the National Aeronautics and Space Administration (80NM0018D0004).

## Author contributions

J. Sauder wrote initially sections I, II, IIIA, IIID, IV,V,and VI. C. Gebara wrote sections IIIB and IIIC. N. Reddy provided extensive rewriting to section III for clarity, and helped the authors find many additional references. C. Garcia-Mora created Figs. 1 and 2 from his extensive experience in working with deployables, contributed to the pyro-cutter section in III.B3, and provided references where these have been used on Small Satellites. All authors extensively edited and revised the sections, and many of the perspectives here reflect their discussions in the process of writing the paper. All authors contributed to references for all sections.

## Competing interests

The authors declare no competing interests.
