## [Peer Review File · Communications Engineering]

Reviewers' comments:

Reviewer #1 (Remarks to the Author):

This manuscript consider functional requirements, design considerations and a qualitative perspective of existing solutions for deployable structures for small satellites. The categorisation into different operational phases (and functional requirements) of a deployable structure is useful, but also follows standard practise in the field. I also struggle with the words 'ontology' and 'ontological' in this context, and would question the correctness of its use here. In discussing examples of different technical solutions, a lot of background knowledge is expected; diagrams and schematics explaining different concepts would have been helpful. The discussion of advantages/disadvantages is necessarily qualitative. Although the observations are generally sensible, no significant new insights or critical views are provided. I could provide additional discussion points on specific categories, but ultimately any categorisation will have edge cases and many observations remain subjective.

The article also focuses on 'small satellites' but does not really explain the specific challenges, other than "these mechanisms must be specifically tailored to reduce complexity". Why is that? Reducing complexity is a general engineering objective, not limited to small satellites. Aspects such as small satellites being highly volume constrained, designers often being constrained by lack of experience and expertise, limited testing facilities, etc. would have set the scene better, and would have enabled observations and recommendations more specifically focused at deployables for small satellites.

Further, this paper sits somewhere between a Review and a Perspective (as the authors themselves indicate in the abstract) and it does not really meet expectations for either. As a review it requires too much background knowledge (although "reading of the references is encouraged") to appreciate the implications of the various concepts (schematics illustrating some selected concepts would have helped), and it does not really provide a critical view of the state of the art in the field. As a Perspective, there is little new insight offered to those experienced in the field, and even for those entering the field it is not clear what exactly the challenges are for deployables for small satellites (aside from "specifically tailored to reduce complexity") and what perspective for the future are offered.

If this work published in the AIAA SciTech conference or even in AIAA Journal of Spacecraft and Rockets, it would potentially serve as a useful reference for a specialised audience; I would provide it as additional reading in my spacecraft deployable structures lectures. It is also useful to support initial design of such deployable structures. However, I do not see it fit with the scope of this journal.

Reviewer #2 (Remarks to the Author):

This paper provides a comprehensive review and perspective on deployable structure approaches for small satellites. Approaches for four stages, stow, restrain, actuate, and locate, are covered, and the advantages, disadvantages, as well as possible failure mechanisms, are discussed. The authors also addressed the design practices. Writing such a perspective recognizes the authors' comprehensive understanding of the application of deployable structures on small satellites. This perspective is of great

guidance for structural design and deployment and stowage schemes for deployable satellites and is in line with the topic of this special issue of communication engineering. However, at present, this paper is insufficiently readable, especially for non-experts in the field or scholars and students new to the field. This perspective is more like an outline, lacking the necessary supporting descriptions. As a result, this paper cannot be accepted in its current form, and major revision is required by considering the following suggestions.

1. Figure 1 provides schematic plots for the approaches in different stages, which is good for this perspective. However, now for each approach, the author explains the basic principles in only 2~3 sentences, which is not enough to understand. I suggest the authors add a specific diagram for each approach, which can be drawn by the authors or cited from others. Combine the diagrams for all the approaches to form a comprehensive diagram, which corresponds to Figure 1. This will greatly enhance the quality and information of the paper and will help readers new to the field to easily visualize and understand each approach.
2. For existing design practices, I would likewise recommend authors add a diagram for each major practice to make it easier for the reader to understand. Similarly, please combine such diagrams into one.
3. Please add a section on challenges and future research directions. Without this section, this perspective seems to have lost its soul.

Dear Reviewers,

The authors wish to greatly thank you for your time and helpful feedback, which we have been able to leverage to improve the paper. Attached you will find an edited manuscript with major revisions (both a version with tracked changes and no changes tracked), and comments in-line below about how we implemented your suggested changes (in italics). To help with these major revisions, and making the paper easier to understand, we have enlisted two additional authors from JPL, but who are new to JPL and from different backgrounds, bringing a more diverse set of eyes on how to improve this manuscript.

Reviewer #1 (Remarks to the Author):

This manuscript consider functional requirements, design considerations and a qualitative perspective of existing solutions for deployable structures for small satellites. The categorisation into different operational phases (and functional requirements) of a deployable structure is useful, but also follows standard practice in the field. I also struggle with the words 'ontology' and 'ontological' in this context, and would question the correctness of its use here.

Thank you for this input. To clarify the text, we have removed references to ontology, and instead just reference this paper as setting up a framework of deployment stages, and approaches to each stage.

In discussing examples of different technical solutions, a lot of background knowledge is expected; diagrams and schematics explaining different concepts would have been helpful. The discussion of advantages/disadvantages is necessarily qualitative. Although the observations are generally sensible, no significant new insights or critical views are provided. I could provide additional discussion points on specific categories, but ultimately any categorisation will have edge cases and many observations remain subjective.

The article also focuses on 'small satellites' but does not really explain the specific challenges, other than "these mechanisms must be specifically tailored to reduce complexity". Why is that? Reducing complexity is a general engineering objective, not limited to small satellites. Aspects such as small satellites being highly volume constrained, designers often being constrained by lack of experience and expertise, limited testing facilities, etc. would have set the scene better, and would have enabled observations and recommendations more specifically focused at deployables for small satellites.

Very good point, I have highlighted these concerns, and several others in the introduction, which hopefully highlights the need especially within small satellites to reduce complexity and increase mechanism robustness.

Further, this paper sits somewhere between a Review and a Perspective (as the authors themselves indicate in the abstract) and it does not really meet expectations for either. As a review it requires too much background knowledge (although "reading of the references is encouraged") to appreciate the implications of the various concepts (schematics illustrating some selected concepts would have helped), and it does not really provide a critical view of the state of the art in the field. As a Perspective, there is little new insight offered to those experienced in the field, and even for those entering the field it is not clear what exactly the challenges are for deployables for small satellites (aside from "specifically tailored to reduce complexity") and what perspective for the future are offered.

From the feedback from the second reviewer, we have added a challenges and future direction section. We have also expanded the literature and examples included to improve the review aspects of the paper, to hopefully provide a more encompassing review, strengthening both the review and perspective aspects. We have also added significantly improved graphics to help may the paper more understandable to someone who is less experienced, with a unique step by step graphic for each approach and design principle.

If this work published in the AIAA SciTech conference or even in AIAA Journal of Spacecraft and Rockets, it would potentially serve as a useful reference for a specialised audience; I would provide it as additional reading in my spacecraft deployable structures lectures. It is also useful to support initial design of such deployable structures. However, I do not see it fit with the scope of this journal.

Thank you for this feedback. We hope the updated graphics will aid your students in understanding the paper content.

Reviewer #2 (Remarks to the Author):

This paper provides a comprehensive review and perspective on deployable structure approaches for small satellites. Approaches for four stages, stow, restrain, actuate, and locate, are covered, and the advantages, disadvantages, as well as possible failure mechanisms, are discussed. The authors also addressed the design practices. Writing such a perspective recognizes the authors' comprehensive understanding of the application of deployable structures on small satellites. This perspective is of great guidance for structural design and deployment and stowage schemes for deployable satellites and is in line with the topic of this special issue of communication engineering. However, at present, this paper is insufficiently readable, especially for non-experts in the field or scholars and students new to the field. This perspective is more like an outline, lacking the necessary supporting descriptions. As a result, this paper cannot be accepted in its current form, and major revision is required by considering the following suggestions.

1. Figure 1 provides schematic plots for the approaches in different stages, which is good for this perspective. However, now for each approach, the author explains the basic principles in only 2~3 sentences, which is not enough to understand. I suggest the authors add a specific diagram for each approach, which can be drawn by the authors or cited from others. Combine the diagrams for all the approaches to form a comprehensive diagram, which corresponds to Figure 1. This will greatly enhance the quality and information of the paper and will help readers new to the field to easily visualize and understand each approach.

Thank you for this very good suggestion. We have created a much better diagram to help visualize each approach.

2. For existing design practices, I would likewise recommend authors add a diagram for each major practice to make it easier for the reader to understand. Similarly, please combine such diagrams into one.

Thank you for this suggestion as well, we have added a second figure as suggested for the design approaches.

3. Please add a section on challenges and future research directions. Without this section, this perspective seems to have lost its soul.

Thank you again for this feedback. We have added a section on challenges and future directions, where first we list three challenges, and then three future directions, or “new opportunities” which exist to resolve each challenge respectively. I greatly appreciate the clear instructions for improving the paper.

REVIEWERS' COMMENTS:

Reviewer #2 (Remarks to the Author):

The authors have well addressed my comments, and I therefore, suggest acceptance of this manuscript as a perspective article.

REVIEWERS' COMMENTS:

Reviewer #1:

None

Reviewer #2:

Remarks to the Author:

The authors have well addressed my comments, and I therefore, suggest acceptance of this manuscript as a perspective article.

AUTHOR NOTES:

We appreciate the reviewers comments, and glad we have answered their requests to their satisfaction. We thank the reviewers for their assistance in improving the paper.